# Decoding the Complexity of Immune–Cancer Cell Interactions: Empowering the Future of Cancer Immunotherapy

**DOI:** 10.3390/cancers15164188

**Published:** 2023-08-21

**Authors:** Kaitlyn Maffuid, Yanguang Cao

**Affiliations:** 1Division of Pharmacotherapy and Experimental Therapeutics, School of Pharmacy, University of North Carolina at Chapel Hill, Chapel Hill, NC 27599, USA; kmaffuid@email.unc.edu; 2Lineberger Comprehensive Cancer Center, School of Medicine, University of North Carolina at Chapel Hill, Chapel Hill, NC 27599, USA

**Keywords:** immunotherapy, cell–cell interaction, intercellular labeling, intercellular imaging, bioinformatics, cancer

## Abstract

**Simple Summary:**

Cell-to-cell communication between the immune system and tumors is of the utmost importance; it influences the development of tumors, their growth, and how they respond to treatments. In this article, we provide an overview of why understanding the interactions between immune and tumor cells is so significant for developing anti-cancer therapeutics, particularly cancer immunotherapy. We delve into the methods and tools used to decipher these interactions and discuss the potential impact on the future of cancer treatment. Moreover, we emphasize the power of unraveling these interactions in advancing cancer immunotherapy. We also explore the challenges that can be tackled by gaining insights into these interactions.

**Abstract:**

The tumor and tumor microenvironment (TME) consist of a complex network of cells, including malignant, immune, fibroblast, and vascular cells, which communicate with each other. Disruptions in cell–cell communication within the TME, caused by a multitude of extrinsic and intrinsic factors, can contribute to tumorigenesis, hinder the host immune system, and enable tumor evasion. Understanding and addressing intercellular miscommunications in the TME are vital for combating these processes. The effectiveness of immunotherapy and the heterogeneous response observed among patients can be attributed to the intricate cellular communication between immune cells and cancer cells. To unravel these interactions, various experimental, statistical, and computational techniques have been developed. These include ligand–receptor analysis, intercellular proximity labeling approaches, and imaging-based methods, which provide insights into the distorted cell–cell interactions within the TME. By characterizing these interactions, we can enhance the design of cancer immunotherapy strategies. In this review, we present recent advancements in the field of mapping intercellular communication, with a particular focus on immune–tumor cellular interactions. By modeling these interactions, we can identify critical factors and develop strategies to improve immunotherapy response and overcome treatment resistance.

## 1. Introduction

Intercellular interactions play crucial roles in organism function and development. Cells can interact with each other either directly (in physical proximity) or indirectly (paracrine signaling). These interactions are the basic building blocks of physiological communication and are essential for tissue formation, immune response, homeostasis, and regeneration. In direct cellular communications, contact between cell surfaces can occur via gap junctions, cell adhesion, tunnel nanotubes, and ligand receptor signaling. When cells interact indirectly, cellular information is shared through signaling from extracellular vesicles, cytokines, chemokines, growth factors, metabolites, and exosomes. These intercellular interacting mechanisms contribute to tissue development and physiological functions [1]. 

A healthy immune system can precisely identify and eliminate precancerous cells, and it eliminates them before they cause any harm, by a process referred to as tumor immune surveillance. Numerous extrinsic and intrinsic factors impair immune–precancerous cell interactions, contributing to tumorigenesis. Once developed, tumor cells evade and disrupt the host immune system, leading to an immune-suppressive tumor microenvironment (TME). The TME is a complex ecosystem comprising tumor cells, immune cells, fibroblasts, extracellular matrices, and signaling molecules. The interaction between the immune cells and cancer cells within the TME evolves and can result in either pro- or anti-tumorigeneses [2]. Restoring the immune function and the network of healthy cell–cell communication within the TME has become a significant component of cancer immunotherapy.

Cancer immunotherapy began in the 1980s with an interferon-α2 inhibitor as the first immunotherapeutic agent approved by the FDA in 1986 [3]. Since then, numerous other immunotherapies have been discovered, including immune checkpoint inhibitors, oncolytic viruses, bispecific T cell engagers, cytokine therapies, and adoptive cell therapies. The central concept of immunotherapy is to restore or reactivate the host anti-tumor immune system [3]. As promising as these immunotherapies are, clinical response varies significantly from patient to patient, primarily because of distinct patterns of immunosuppressive TMEs and different patterns of disruption in cell–cell interactions. About 30–40% of patients respond to immunotherapy, with fewer achieving a durable response [4]. Variability in the TME primarily relates to the different degrees of tumor-infiltrated lymphocytes (TILs) and their functions. Some patients have “hot” tumors, where the tumors have higher TILs, and these patients usually respond well to immunotherapy [5]. On the other hand, some patients have “cold” tumors with poor or almost no TILs, and these tumors often develop resistance to immunotherapy [5]. Elucidating the mechanism of resistance and characterizing the distorted patterns of intercellular interactions between tumor and immune cells within TMEs has become critical for more potent immunotherapies.

Studies over the years have showcased the cell types present in the TMEs associated with positive and negative outcomes of immunotherapy. High CD8^+^ T cell abundance is often associated with favorable overall survival, whereas increased regulatory T cells are associated with poor overall survival [6,7]. However, a recent pan-cancer analysis showed that high CD8^+^ abundance is not always associated with a better prognosis, as the spatial cellular assemblies are also crucial [8], which means that different immune cells will have various prognostic factors depending on the location and type of cancer. In addition to PD-L1 expression and tumor mutation burdens, there are still no robust prognostic biomarkers across cancers. 

Methods that can characterize the deformed intercellular interactions and identify the cell (sub-) populations that are involved in the interactions are extremely critical for prognosis. Identifying which cell population at which cellular state is associated with positive or negative outcomes in immunotherapy is extremely valuable to patient stratification, prognostic biomarkers, and resistance mechanisms. This review aims to explore the diverse manifestations of cell–cell interactions in the context of cancer, providing an overview of quantitative approaches to assess these interactions. Moreover, it seeks to investigate the potential of leveraging the influence of cell–cell interactions to enhance the efficacy of cancer immunotherapies.

## 2. Cell–Cell Interactions during Tumorigenesis

The immune system consists of two compartments: innate cells (such as macrophages, neutrophils, dendritic cells, and natural killer cells) and adaptive cells (B cells and T cells). Innate cells rapidly respond to foreign pathogens, presenting antigens to adaptive cells to initiate specific immunological responses. In the context of cancer, antigen-presenting cells detect tumor antigens and present them to naïve lymphocytes. This communication primes and activates lymphocytes, which then migrate to the tumor site. The activated T cells recognize and eliminate tumor cells. The innate and adaptive immune cells collaborate in a process called cancer immunoediting to eliminate tumors. However, if this process is unsuccessful or suppressed, the tumor microenvironment forms. 

The progression of the TME towards malignancy can be understood within the framework of cancer immunoediting. Cancer immunoediting refers to the dynamic interplay between the host immune system and the tumor cells, whereby immune mechanisms either restrain or promote tumor development. This process can be divided into three distinct phases: elimination, equilibrium, and escape [9].

During the elimination phase, innate and adaptive immune responses collaborate to recognize and eliminate malignant cells. The innate immune system, including dendritic cells and antigen-presenting cells, primes and activates T cells by presenting tumor antigens. These activated T cells are then mobilized to directly interact with cancer cells, leading to their destruction [9]. However, tumors can evade elimination by exploiting immune checkpoints, such as PD-1/PD-L1 and CTLA-4, which act as brakes on T cell activity. When T cells engage with cancer cells bearing PD-L1 or CTLA-4, inhibitory signals are transmitted, causing T cell exhaustion and inactivation [9]. 

Tumors that successfully evade elimination enter the equilibrium phase, during which they remain dormant but develop resistance mechanisms against immune surveillance [9]. This resistance is mediated by immunosuppressive cell types, including tumor-associated macrophages (TAMs), myeloid-derived suppressor cells (MDSCs), cancer-associated fibroblasts (CAFs), and regulatory T cells (Tregs) [10]. These cells actively communicate with other components of the TME, dampening T cell activation, modulating effector cell function, and promoting tumor progression. Of special significance, CAFs and TAMs play pivotal roles in carcinogenesis and the maturation of TMEs [10]. CAFs can promote tumor growth, angiogenesis, the invasion of tumor cells into surrounding tissues, and the modulation of the tumor response to immunotherapy [11,12,13]. They also secrete various signaling molecules and cytokines that can modulate immune responses and create an environment favorable for tumor growth. The equilibrium phase sets the stage for the subsequent escape phase, characterized by clinically detectable tumor growth and the need for therapeutic intervention [9,10].

Effective immunotherapy can revert tumors to the elimination phase, where the suppressive mechanisms are counteracted, leading to the elimination of the tumor [9]. However, a partial response to immunotherapy may shift the tumor back into the equilibrium phase, enabling the emergence of resistant clones and eventually leading to the escape phase, signifying acquired resistance to immunotherapy [9]. In cases where immunotherapy fails to induce a response, the tumor demonstrates innate resistance to treatment [9].

Understanding the intricate dynamics of cancer immunoediting and intercellular interactions between tumor and immune cells is crucial for the development of effective therapeutic approaches aimed at restoring immune control over tumors and achieving durable clinical responses.

## 3. Cell–Cell Interactions Are the Pharmacological Basis of Immunotherapy

Almost all types of immunotherapies involve direct cell–cell interactions for anti-tumor effect. One of the key cell–cell interactions during immunotherapy is the interaction between T cells and tumor cells. T cells can recognize and target tumor cells through the recognition of specific antigens presented by the tumor cells. However, tumors can evade T cell recognition by downregulating antigen presentation or by producing immune-suppressive molecules. Cell–cell interactions are the pharmacological basis of immunotherapy (Figure 1). The most successful immunotherapy—immune checkpoint inhibitors—have been approved for over 19 types of cancer treatment [14]. Blocking the immune checkpoint, CTLA-4 or PD-1/PD-L1, can restore the function of TILs for cytotoxic effects, which entail direct physical and functional contact between TILs and tumor cells. TILs, once engaged with target cells, can secrete perforin and granzyme B for cytotoxic effects [15]. 

Another mechanism in which immune cells can have a cytotoxic effect on cancer cells is the process of antibody-dependent cellular cytotoxicity (ADCC) [15,16]. In ADCC, tumor-specific monoclonal antibodies (mAbs) recognize tumor-selective antigens on the surface of cancer cells. The Fc receptor expressed by the effector immune cell binds the Fc portion of the antibody attached to the cancer cells. Upon binding, the immune cell secretes proteins and enzymes, inducing cancer cell lysis. Many IgG-based targeted therapies, such as rituximab and trastuzumab, can trigger antibody-dependent cellular cytotoxicity (ADCC) through interactions between Fc and Fcγ receptors expressed on effector cells, initiating direct cell–cell interactions and cytotoxicity [16]. 

Another therapeutic approach that utilizes antibodies is bispecific T cell engagers (BiTEs) [17]. BiTEs are characterized by having two different antigen-binding sites in a single molecule, with one site binding to T cell receptors to activate cytotoxic T lymphocytes, and the other site binding to tumor-specific antigens (TSAs). The engagement between cytotoxic T lymphocytes and tumor cells triggered by BiTEs leads to the elimination of the tumor cells. Examples of BiTEs include CD3 and 4-1BB, which activate cytotoxic T lymphocytes, and target tumor-associated antigens (TAA), such as CD19 and CD20. BiTEs redirect cytotoxic T lymphocytes to specifically recognize and engage tumor cells, initiating cell–cell contact, known as the immunological synapse, and inducing cytotoxicity [17,18]. 

Chimeric antigen receptor (CAR) cell therapies represent a novel immunotherapeutic approach that signifies a significant advancement in personalized cancer treatment [19]. This approach involves genetically modifying T cells or natural killer (NK) cells to express synthetic receptors (CARs) that can bind to tumor antigens [19]. This genetic modification enables the redirected T or NK cells to specifically recognize cancer cells and initiate immune responses against them. 

Oncolytic virus therapy holds promise as an immunotherapy approach that involves T cell activation and cell–cell interactions. This therapy utilizes either genetically engineered or naturally occurring viruses that can selectively replicate within cancer cells and kill them while sparing non-cancerous cells [20,21]. Upon administration, the oncolytic virus activates the immune system, leading to the recruitment of natural killer (NK) cells and CD8^+^ T cells to the tumor site. This process results in the reduction in regulatory T cells (T_regs_) and facilitates an effective immune response against the cancer cells [20,21].

In summary, intercellular interactions between effector cells (such as cytotoxic T lymphocytes and NK cells) and tumor cells have emerged as crucial steps for the efficacy of immunotherapies. Therapeutic approaches such as BiTEs, CAR cell therapies, and oncolytic virus therapies exploit these interactions to enhance the immune response against cancer cells, and they hold promise for the improvement of cancer treatment outcomes.

## 4. Experimental and Modeling Systems for Studying Cell–Cell Interactions

The intricate interplay between immune cells and structural components within the TME significantly influences patient outcomes, therapeutic response, and disease progression [22]. While existing data shed light on the communicative relationships between immune cells and tumor cells within the TME, there remains a substantial knowledge gap with regard to the intra-patient and intra-cancer types of communication. Consequently, the TME and its diverse cellular components have emerged as an enticing landscape for the research aiming to discover novel therapeutic strategies and optimize patient management [22].

To investigate and elucidate the intricate cellular interactions within the TME, numerous methodologies and systems have been developed (Figure 2). These approaches encompass in vitro and in vivo models, employing molecular analysis techniques, proximity labeling methods, and bioinformatic approaches [22,23]. Understanding the composition of distinct cell types within different TMEs and patient contexts is pivotal for advancing therapeutic interventions and identifying prognostic biomarkers. In this section, we delve into various experimental techniques and modeling systems for the comprehensive study of cell–cell interactions, encompassing experimental modeling systems, microscopy/imaging, proximity labeling, and bioinformatic approaches.

The study of cell–cell interactions employs various in vitro experimental systems, including two-dimensional (2D) cell culture and three-dimensional (3D) methods such as spheroids and organoids, as well as tissue samples (Table 1). Traditional 2D culture using primary cells and cell lines has long been considered a gold standard in cell culture due to its cost-effectiveness, long-term culture viability, low maintenance requirements, and user-friendly nature [24]. However, 2D culture falls short in mimicking the natural tissue structure, and it lacks biologically relevant cell–environment interactions when investigating complex environments like TME or normal tissue structures. To address this limitation, 3D cell culture techniques have revolutionized in vitro methodologies by providing more physiologically relevant options [24]. Organoids and spheroids have gained popularity as they enable the comparison of in vivo organs in vitro. Organoids, derived from stem cells or patient tumor cells, are three-dimensional tissue cultures that replicate the morphological and genetic features of the original tumor, allowing for patient-specific models and in vitro representations of the TME [25]. However, organoids have limitations, such as high patient variability, absence of specific essential cellular components, challenging culture maintenance, and higher costs [25]. On the other hand, spheroids are simpler three-dimensional clusters of cells derived from various cell types, including tumor tissues and hepatocytes [26]. They do not require scaffolding to form 3D cultures but rely on cell adhesion. However, spheroids lack the ability to self-assemble or regenerate, making them less desirable compared to organoids [26]. Both models enable three-dimensional assessment of tumors in vitro, providing improved translational models for clinical applications. Tissue biopsy slices are also valuable for identifying the spatial distribution and location of cells within the TME or normal tissues. However, the slicing process introduces variability in cell distribution due to the method employed [27].

Nevertheless, in vitro systems lack the host tissue contexture and immune system, which are critical components for studying cell–cell interactions. Cell–cell interactions rely heavily on the tissue contexture, and techniques that support in vivo investigations have the potential to unveil novel modes of cell–cell interactions and their impact on tumor response to therapies. In vivo systems with an intact host immune system, such as syngeneic mouse models, are useful for studying the TME since the host immune system can interact with the TME [28,29]. However, a drawback of in vivo models lies in the fact that the interacting immune system being studied is often the host mouse immune system, which differs significantly from the human immune system, especially concerning cellular interactions [28].

Furthermore, tumor xenograft models using cell lines often undergo substantial genetic changes, fail to recapitulate the natural tumor structure, and may lead to mouse-specific tumor evolution [29]. While mice are the preferred experimental model for immunologists, there are significant differences between mice and humans, particularly in innate and adaptive immunity, leading to challenges in translating findings to humans [30].The low success rate of clinical trials, which is less than 15%, can be attributed, in part, to the inadequate modeling of human diseases in animals and the limited predictability of animal models [31]. Future advancements in ex vivo models and platforms, such as microfluidics, hold promise with regard to the use of patient-derived human samples to study cell–cell interactions, leading to better clinical translation. 

## 5. Proximity-Based Labeling Approaches for Studying Cell–Cell Interactions

Intercellular proximity labeling approaches have revolutionized the study of cell–cell interactions by providing spatially resolved information. These methods involve the tagging or labeling of proteins or other molecules that are in proximity to a specific cell type or surface marker [32] (Table 2). By identifying and analyzing the labeled molecules, researchers can gain valuable insights into the neighboring cell types and their interactions. In the context of studying the TME, proximity labeling approaches play a crucial role. Understanding the interactions between cancer cells and immune cells within the TME is essential for developing effective immunotherapies. By labeling and tracking immune cells that have come into proximity or contact with tumor cells, we can investigate the molecular features of immune cells and assess how these cells influence the composition and function of the TME.

Unlike the IHC or fluorescent staining approaches, proximity labeling techniques can go beyond a time-frozen snapshot, helping to identify and track cells and providing a more dynamic approach to cell–cell interactions. This becomes important when studying where immune cells go after interacting with cancer cells and how that cellular movement affects the composition of the TME. To ensure the applicability of proximity labeling approaches in both in vivo and in vitro environments, it is essential that these methods are non-disruptive and non-toxic to cells. This consideration ensures that the labeled cells maintain their physiological properties and behave naturally during the experimental process. By utilizing labeling techniques that are minimally invasive and compatible with live cell imaging, researchers can gain a comprehensive understanding of cell–cell interactions in the TME.

Two prominent intercellular proximity labeling methods, EXCELL and LIPSTIC, employ the Staphylococcus aureus enzyme Sortase A (SrtA) to measure cell–cell interactions. LIPSTIC (Labeling Immune Partnerships by SorTagging Intercellular Contacts) enables the identification of ligand–receptor interactions between immune cells and their target cells [33,35]. Cells expressing SrtA on their surface covalently attach biotin molecules to neighboring surface proteins upon cell–cell contact [45]. The interacting cells are then exposed to a streptavidin-conjugated fluorescent dye, allowing for quantification of the interaction. LIPSTIC offers an unbiased approach for identifying ligand–receptor interactions and allows for the study of interaction dynamics over time. However, it relies on the genetic modification and the expression of both donor and receiver cells, limiting its application to specific cell types or tissues [35]. EXCELL (enzyme-mediated intercellular proximity labeling) represents a recent development; it uses an SrtA variant, mgSrtA, enabling promiscuous labeling of various cell surface proteins containing a monoglycine residue at the N-terminus [33]. Unlike LIPSTIC, EXCELL does not require genetic modification of the receiver cells, and it supports the identification of novel cellular interactions, including the subtype identification of TILs interacting with tumor cells. For both approaches, the biotin-labeled proteins can then be isolated and identified using streptavidin-based purification methods such as flow cytometry, and these labeled cells could then be subjected to further molecular characterization.

GFP-based Touching Nexus or G-baToN harnesses the trogocytosis communication of cells to transfer GFP from a donor cell to an acceptor cell [34]. However, the G-baToN approach requires the donor and receiver cells to both be transfected, which does not support the identification of unknown or novel CCIs [34]. FucoID has several advantages over other proximity labeling approaches. It enables the labeling of glycoproteins, which are an important class of proteins involved in many biological processes [36,37]. Additionally, the fucose tag is relatively small, and it interferes minimally with the function of the labeled proteins, which can reduce the likelihood of introducing artifacts into downstream analyses. FucoID can also be combined with other techniques, such as single-cell RNA sequencing, to gain a deeper understanding of the molecular mechanisms underlying intercellular communication. However, FucoID also has some limitations. It is dependent on the expression level and accessibility of the cell surface marker of interest, which may limit its application to certain cell types or tissues. Additionally, the labeling efficiency of FucoID may be affected by the density of glycoproteins on the cell surface and the availability of fucose residues. Careful experimental design and validation are necessary to ensure the accuracy and specificity of the results obtained using FucoID [36,37].

A subsequent method that is able to detect membrane proteins through proximity labeling is population-based interaction tagging (PUP-IT) [38]. In this system, the small protein tag Pup is weakly attached to proteins or prey interacting with the gene *PafA* or bait [38]. PUP-IT was utilized to label the interaction between CD28-expressing Jurkat T cells and CD80/86-expressing Raji B lymphocytes. PUP-IT CD28 extracellular Jurkat T cells were able to label the Raji B cell in vitro [38]. However, for this interaction to be observed both cells needed to be modified with the prey or bait genes. PUP-IT is also classified as a “weak” interaction and may not be suitable for long-term tracking [38]. This highlights that PUP-ID needs some knowledge of ligand–receptor interactions before using. 

2CT-CRISPR assay is a novel and interesting approach for identifying genes that are essential for effector T cell function in tumors. In the 2CT-CRISPR assay, human T cells were represented as effectors, and melanoma cells were represented as targets [39]. The purpose of this assay was to determine whether genetically manipulating the immune cell would influence the tumor cell during ligand–receptor interactions [39]. A recombinant TCR-engineered CD8^+^ T cell was used to target a specific antigen (NY-ESO-1) that can mediate tumor size in melanoma patients. The 2CT method was used to control the selection pressure and killing effects shown by the T cell as well as to modulate the effector to target ratio. Furthermore, the 2CT method was used in combination with a CRISPR-Cas9 library that held over 100,000 single-guide RNAs which impaired effector function in T cells. The 2CT method allowed for the analysis of genes necessary for immunotherapy, specifically those that target effector T cell function. The 2CT method has exciting translation and clinical opportunities to uncover genes in immunotherapy-resistant patients [39]. 

Other proximity-based methods for studying cell–cell interactions include TRACC (Transcriptional Readout Activated by Cell–Cell Contacts) and SynNotch-activated MRI (magnetic resonance imaging), both of which exploit specific receptor–ligand interactions between two interacting cells to facilitate labeling and detection [40,43]. TRACC utilizes a g-protein-coupled receptor with a light-sensitive domain to detect cell–cell interactions using a transcriptional readout [40]. This approach allows for the visualization and identification of cell populations involved in the interaction of interest. SynNotch-activated MRI combines synthetic biology and imaging techniques to detect cell–cell interactions [43]. It involves the engineering of cells expressing a synthetic Notch receptor that can be activated upon interaction with a specific ligand presented by neighboring cells. Upon activation, the engineered cells produce a contrast agent detectable by MRI, enabling the visualization and tracking of the interacting cell populations [43].

Most proximity-based methods are dependent on cell–cell contact and interaction. Cherry-niche, caged luciferins, and CLIP (cre-induced intercellular labeling protein) are three approaches that do not solely rely on direct cell–cell contact for labeling [41,42,44]. In the Cherry-niche method, cells are engineered to express the enzyme Cherry-tagged ligase, which can attach a fluorophore to nearby cells expressing a complementary Cherry-tagged receptor [41]. This proximity labeling occurs within a specific microenvironment or niche defined by the presence of the ligase and receptor [41]. Caged luciferins, on the other hand, involve the use of caged luciferin molecules that can be activated by specific enzymes or stimuli produced by engineered cells [42]. Upon activation, the caged luciferins produce luminescent signals that can be detected and used to identify neighboring cells in the vicinity [42]. CLIP has an interesting methodology in that it can label cells that are in direct contact as well as those that are not in direct contact but are in proximity to each other [44]. This method involves the engineering of both the donor and receiver cells, where the donor cell secretes a lipid-soluble tag containing mCherry that labels the recipient cells [44]. These approaches provide additional tools for studying cell–cell interactions, offering different mechanisms for labeling and detection beyond direct cell–cell contact.

In summary, intercellular proximity labeling or imaging approaches offer significant potential for elucidating the intricate cellular interactions and communication networks within the TME. These methods have been successfully employed in various research areas, including the investigation of tumor metastasis [37,41], T cell priming [35], cell migration [35], tumor–immune cell interactions [37], cellular therapy [37], and the examination of interactions between neurons and glioma cells [40]. As these techniques continue to advance, it is anticipated that their application will expand further, enabling a broader understanding of the factors and mechanisms that impede the pharmacological effects of immunotherapies. Overall, intercellular proximity labeling approaches can provide valuable insights into the complex cellular interactions and communication networks in the TME, which can inform the development of more effective cancer immunotherapies. 

## 6. Bioinformatic Techniques for Inferring Cell–Cell Interactions

Enzyme-based intercellular proximity labeling approaches are predominantly employed in experimental systems. However, with the growing availability of large clinical datasets, bioinformatic methods have gained significance in the study of cell–cell interactions and the identification of novel interactions. In clinical settings, bioinformatics methods play a crucial role in inferring intercellular interactions or communication by examining the coordinated expression patterns of ligand–receptor pairs’ cognate genes. Ligand–receptor analysis has emerged as a valuable approach for investigating intercellular communication, particularly in the context of cancer immunotherapy. This approach enables the identification of the specific ligand–receptor pairs involved in immune cell interactions with cancer cells or the TME (Table 3). 

This approach proves to be especially valuable in deducing intercellular interactions that are not solely reliant on cell-to-cell contact. This is evident when immune cells and cancer cells release diverse cytokines, chemokines, and growth factors that govern immune reactions and inflammation. These signaling molecules and their corresponding receptors may be modulated based on environmental cues. Through these bioinformatics methods, we can deduce the likelihood of intercellular interactions based on their ligand–receptor profiles; this is potentially pivotal in forecasting patient prognosis and treatment outcomes.

The analysis of ligand–receptor interactions primarily relies on single-cell RNA sequencing (scRNA-seq) or bulk RNA-seq data. The procedure typically involves the following steps:

Data preprocessing: This step involves normalizing, quality controlling, and filtering of gene expression data to ensure data integrity and reliability.

Gene set selection: A specific set of ligand and receptor genes is chosen based on prior knowledge or by utilizing databases such as CellPhoneDB or Interactome INSIDER.

Calculation of ligand–receptor expression: The expression levels of ligand and receptor genes are calculated for each cell or cell type within the dataset, regardless of whether the data are scRNA-seq or bulk RNA-seq. 

Ligand–receptor interaction analysis: Interactions between ligands and receptors are predicted by assessing the co-expression patterns of ligand and receptor genes across different cells or cell types. Several methods, including CellPhoneDB, scRNA-seq-based ligand–receptor pair analysis (sLRPA), and LigandNet, are available for performing this analysis. 

Visualization and interpretation: The results of the analysis are visualized using heatmaps, networks, or other visualization techniques. These results can be interpreted to identify the specific ligand–receptor pairs that may be involved in intercellular communication and to gain insights into the underlying biological processes.

Bioinformatic methods offer valuable tools for inferring intercellular interactions and communication based on transcriptome data. These approaches provide valuable insights into the intricate network of cell–cell interactions in the context of cancer immunotherapy. Notably, ligand–receptor analysis holds promise for the identification of predictive biomarkers for immunotherapy response and the monitoring of treatment efficacy.

Nevertheless, it is important to acknowledge the limitations of these bioinformatic methods in inferring intercellular interactions. Firstly, their predictions solely rely on gene expression data and overlook additional factors like post-translational modifications or protein localization that can influence interactions. Secondly, these methods hinge upon existing knowledge of ligand–receptor pairs, which may be incomplete or imprecise. Thirdly, the biological relevance of the predictions is not guaranteed, necessitating experimental validation. Lastly, technical artifacts such as batch effects, sequencing depth, and normalization methods can influence the accuracy and reproducibility of the results. Considering these limitations, it is crucial to exercise caution and combine bioinformatic predictions with experimental validation to ensure the reliability and significance of the findings. Continued advancements in bioinformatic techniques and complementary experimental approaches will enhance our understanding of intercellular interactions and their role in cancer immunotherapy. 

## 7. Potential Questions to Be Addressed by These Approaches

A wide array of scientific questions concerning cancer immunotherapy could be answered by utilizing these cell-cell interactions techniques. These questions include:

What are the subpopulations of immune cells interacting with the tumor cell during immunotherapy?

What are the molecular features of these interacting immune cells, and how are the molecular features related in response to immunotherapy? 

Do the effector cells that have interacted with the tumor cells migrate across tumor metastatic lesions? 

Ultimately, the answers to these questions can uncover the pharmacological actions of cancer immunotherapy and reveal the underlying molecular mechanism of resistance. 

## 8. Concluding Remarks

The TME represents an intricate network comprising diverse cell types engaged in communication; it plays a pivotal role in shaping the tumor landscape. Effective communication between immune cells and cancer cells holds great significance in the determination of the patients’ responses to immunotherapy, and it contributes to treatment resistance and interpatient variability in the responses. The investigation of cell–cell communication in immune cells under normal and pathological conditions provides crucial insights into the mechanisms of cancer immunotherapy, patient responses, disease progression, and TME status. Various experimental and computational approaches exist for elucidating pathological intercellular interactions, both directly and indirectly, with the aim of identifying the communicating cell populations. Comprehensive understanding, modeling, and the discovery of cell–cell interactions within the TME hold immense potential for the identification of the critical factors and strategies influencing immunotherapy response, treatment resistance, and TME status.

In recent times, advances in imaging, microscopy, cellular engineering, and bioinformatics have emerged as powerful tools for the unraveling of novel mechanisms and cellular relationships, thereby paving the way for improved immunotherapy options for patients. By leveraging these methodologies synergistically, it becomes possible to bridge existing knowledge gaps and gain a comprehensive understanding of treatment resistance and to design more potent cancer immunotherapies. 

## Figures and Tables

**Figure 1 cancers-15-04188-f001:**
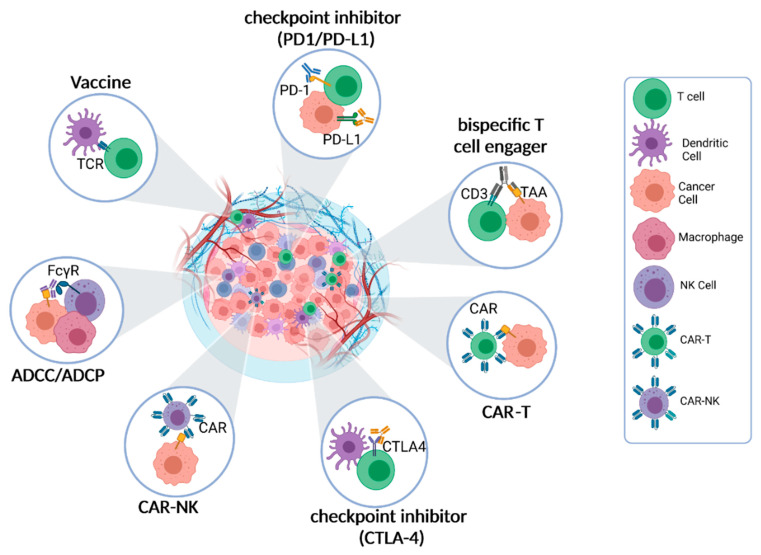
Cell–Cell interactions are the pharmacological basis of immunotherapy. The primary pharmacological mechanism of immunotherapy entails the activation or engagement of diverse immune cell populations to identify and eradicate cancer cells. This crucial process heavily relies on the dynamic interplay between immune cells and cancer cells within the TME. Within the TME, immune cells, including T cells, natural killer (NK) cells, dendritic cells (DCs), and macrophages, establish various types of interactions with cancer cells. These interactions encompass intricate molecular signaling pathways, direct cell-to-cell contact, and the exchange of soluble factors. The ultimate objective of immunotherapeutic approaches is to either alleviate immune suppressive interactions within the TME or activate immune effector functions, thereby unleashing the anti-cancer pharmacological effects of immunotherapy.

**Figure 2 cancers-15-04188-f002:**
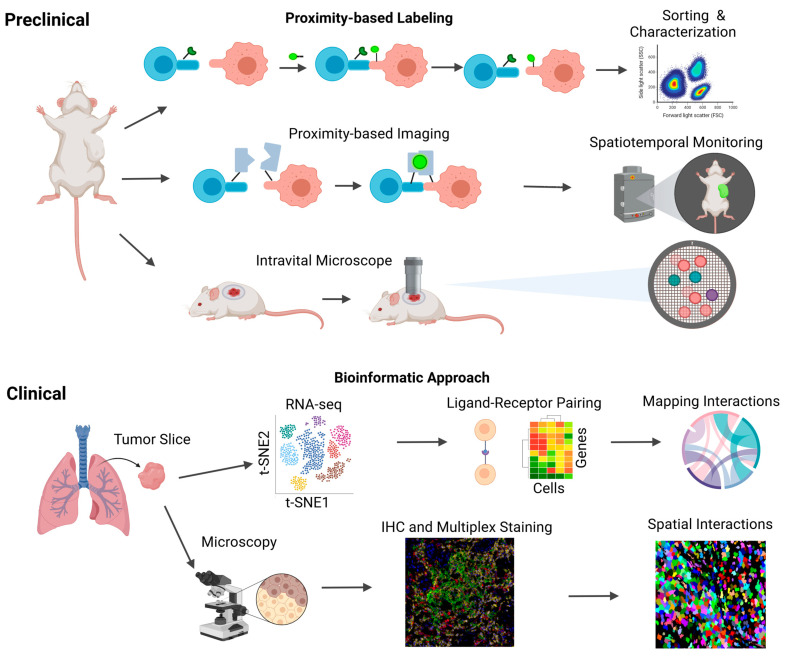
Human and murine methods to study cell–cell interactions. Schematic overview of the current human and murine methods for studying intercellular interactions. Human samples, obtained primarily from tissue biopsies and blood samples, serve as valuable resources for investigating these interactions. Two key techniques employed in human models are single-cell RNA sequencing (scRNA-seq) and immunohistochemical staining (IHC). scRNA-seq enables the analysis of ligand–receptor interactions, facilitating the mapping of diverse cell–cell interactions. On the other hand, IHC provides spatial information, allowing for the identification of cell locations and their physical proximity to one another. In addition to the techniques utilized in human models, other innovative methods have been developed to study intercellular interactions. These include proximity-based intercellular labeling approaches such as LIPSTIC and EXCELL, which enable the identification of neighboring cells and the assessment of their interactions. Proximity-based intercellular imaging approaches such as confocal microscopy and CODEX offer further insights into intercellular communication by visualizing the spatial relationships between cells. Furthermore, intravital microscopy has emerged as a powerful tool for the real-time monitoring of lymphocyte localization and movement within tumor microenvironments.

**Table 1 cancers-15-04188-t001:** Experimental modeling systems for intercellular interactions.

Model System	Sample	Method Description	Reference
2D cell culture	Cells	Cells grow in a monolayer if adherent or suspended in a culture flask. These cultures are a straightforward, cost-effective, and low-maintenance approach. Within the controlled environment, it is possible to investigate the interactions between different cell lines and observe their behavior and responses to treatments.	[24]
3D cell culture	Cells	Cell growth and interactions occur in 3D space, where cells interact with their surrounding environment and neighboring cells. Two approaches: scaffold-based methods using hydrogels or structural scaffolds and scaffold-free techniques (spheroids).	[24]
Spheroids	Cells	Organoids, also known as multicellular spheroids, are self-assembled structures that mimic the physiological environment and interactions found in vivo. They provide a more physiologically relevant context, allowing the investigation of intercellular interactions and responses within a 3D microenvironment resembling in vivo conditions.	[25]
Organoids	Patient-derived cells and tissues	Primary patient-derived microtissues grown in a 3D extracellular matrix that represents in vivo physiology and genetic diversity, allowing the investigation of intercellular interactions and responses in a patient-specific manner.	[26]
Tissue Slices	Tumor Tissue	Tumor biopsy taken from patients or xenograft models, stained to assess tumor morphology and spatial location of cells.	[27]
Animal models	Tumor Tissue	Compatible with intravital and intercellular imaging/labeling techniques, as well as other genetic systems designed to detect cell–cell interactions upon contact or external stimulation, including UV or fluorescent light.	[28]

**Table 2 cancers-15-04188-t002:** Proximity-based labeling approaches for studying cell–cell interactions.

System	Scale	Application	Method	Reference
EXCELL	In vitro	Labeling Imaging	EXCELL (enzyme-mediated intercellular proximity labeling) is a method that utilizes a variant of SrtA, mgSrtA, to enable the non-specific labeling of cell surface proteins containing a monoglycine residue at the N-terminus. Unlike other methods, EXCELL does not require pre-engineering of acceptor cells and was applied in in vitro studies.	[33]
G-BaToN	In vitro In vivo Ex vivo	Labeling Imaging	G-BaToN is a versatile system for physical contact labeling between cells. Sender cells express surface-bound GFP, while receiver cells carry a synthetic element that selectively binds to GFP. Upon cell contact, GFP is transferred from sender to receiver cells, leading to fluorescence labeling of the receiver cells. This method requires pre-engineering of both sender and receiver cells and can be used for in vitro and ex vivo studies.	[34]
LIPSTIC	In vitro	Labeling	LIPSTIC (Labelling Immune Partnerships by SorTagging Intercellular Contacts) is a proximity-dependent labeling method that employs bacterial sortase (SrtA) to detect receptor–ligand interactions between cells. It involves the attachment of biotin to cell surface proteins, which can be detected using flow cytometry. LIPSTIC can be used in both in vitro and in vivo settings by pre-engineering the cells on both sides of the interaction.	[35]
FucoID	In vitro Ex vivo	Labeling	FucoID is a method for identifying antigen-specific T cells using interaction-dependent fucosyl biotinylation. This technique enables the isolation of endogenous tumor antigen T cells from tumor digests without prior knowledge of the tumor-specific antigens and has been used for ex vivo studies.	[36,37]
PUP-IT	In vitro	Labeling	PUP-IT (pupylation-based interaction tagging) is a method used to identify membrane protein interactions. In this approach, a small protein tag, Pup, is applied to proteins that interact with a PafA-fused bait, enabling transient and weak interactions to be enriched and detected by mass spectrometry. PUP-IT enables the identification and analysis of protein–protein interactions occurring at the membrane level.	[38]
2CT-CRISPR	In vitro Ex vivo	Genetic influence	Two-cell type CRISPR assay. This assay can genetically manipulate T cells to interact with cancer cells ex vivo to determine the genes that influence T cell effector function on cancer cells.	[39]
TRACC	In vitro	Labeling Imaging	TRACC (Transcriptional Readout Activated by Cell–Cell Contacts) is a system that utilizes light gating to detect cell–cell contacts based on transcriptional activity (TF). Cells are engineered to express a light-responsive TF that regulates the expression of a reporter gene. When two cells come into contact, a light signal is applied to activate the TF, resulting in the activation of the reporter gene and subsequent detection of the cell–cell contact, monitoring cell–cell interactions in a controlled and dynamic manner.	[40]
Cherry-niche	In vivo	Labeling Imaging	Cherry-niche is an innovative method that allows cells expressing a fluorescent protein to selectively label their surrounding cells in the tumor niche. This technique involves generating cancer cells capable of transferring a liposoluble fluorescent protein to their neighboring cells within the tumor microenvironment.	[41]
Caged luciferins	In vitro In vivo	Imaging	Caged luciferins are utilized for bioluminescent activity-based sensing. Activator cells expressing β-galactosidase catalyze the cleavage of caged luciferin, known as Lugal, resulting in the release of D-luciferin. The liberated D-luciferin can then enter nearby reporter cells, where it serves as a substrate for the luciferase enzyme, leading to the production of light and allowing for the identification and visualization of cells that are in close proximity to the sender cells.	[42]
SynNotch-activated MRI	In vivo	Imaging	The SynNotch system is utilized to induce the expression of an MRI contrast agent in recipient cells when they interact with sender cells expressing the corresponding synthetic notch receptor, enabling the detection and visualization of cell–cell communication events in real time.	[43]
CLIP	In vivo	Labeling Imaging	CLIP (cre-induced intercellular labeling protein) secretes a membrane-permeable fluorescent protein (mCherry) from a donor cell that can mark neighboring receptor cells. This method can label both direct cell contact receptor cells and receptor cells at a close-range distance.	[44]

**Table 3 cancers-15-04188-t003:** Bioinformatic techniques for inferring cell–cell interactions.

Platform	Data Source	Method	Reference
CellTalkDB	scRNA-seq	Manually curated database of ligand–receptor pairs from both human and mouse samples.	[46]
iTalk	scRNA-seq	Identifying and illustrating alterations in intercellular signaling network. R package made to analyze and visualize ligand–receptor pair.	[47]
PyMINeR	scRNA-seq	Python maximal information network exploration resource. Fully automates cell type-specific identification, and pathways as well as in silico detection of autocrine and paracrine signaling networks	[48]
CellChat	scRNA-seq	Open source R package that is able to visualize, analyze, and deduce intercellular communications from a data input. Uses mass action models and differential expression analysis to deduce cell state-specific signaling communications. Also provides visualization outputs to compare intercellular communication methods.	[49]
CellPhoneDB	scRNA-seq	Identifies biologically relevant interacting ligand–receptor pairs. Cells with the same cluster are pooled together as one cell state. Ligand–receptor interactions are derived based on the expression of a receptor of one state and a ligand of the other state.	[50]
Giotto	scRNA-seq	Open source spatial analysis platform that contains two modules, Giotto analyzer and Giotto viewer, which are both independent and fully integrated. Analyzer provides instructions about steps in analyzing single-cell expression data, and the viewer provides an interactive view of the data.	[51]
ICellNET	RNA-seq, scRNA-seq, and microarray	Transcriptomic-based framework that integrates a database of ligand–receptor interactions, communication scores, and connections of cell populations of interest with 31 human reference cell types and three visualization methods.	[52]
SingleCellSignalR	scRNA-seq	Open source R platform. Relies on a database of known ligand–receptor interactions called LR*db*.	[53]
CCC Explorer	Transcriptome profiles	Java-based software. Uses a computational model to look at cell–cell communications ranging from ligand–receptor interactions to transcription factors and target genes.	[54]
NicheNet	Gene expression data	Open source R platform. Uses a database of ligand–receptor interactions to identify ligand–receptor interactions that could drive gene expression changes	[55]
SoptSC	RNA-seq	Similarity matrix-based optimization for single-cell data analysis. Uses a cell-to-cell similarity matrix via gene marker identification, lineage reference, clustering, and pseudo-temporal ordering. From this information, it predicts cell–cell communication networks.	[56]
SpaoTSC	scRNA-seq	Spatially optimal transporting of the single cells. The method has two major components: (1) constructing spatial metric for cells from scRNA-seq data and (2) reconstructing the cell–cell communication networks from the data and identifying relationships between genes from intercellular relationships. Uses python.	[57]
scTensor	scRNA-seq	Open source R package. Instead of looking at one-to-one cell–cell interactions, this software focuses on many-to-many cell–cell interactions. scTensor looks at a three-way relationship (hypergraph) between ligand expression, receptor expression, and ligand–receptor pairs.	[58]

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
