# Peer review of "Decoding the Complexity of Immune–Cancer Cell Interactions: Empowering the Future of Cancer Immunotherapy"

_cancers, 2023, doi:10.3390/cancers15164188_

Round 1

Reviewer 1 Report

Review, in my opinion  is rather well written  and I hope will be useful for readers. 

Some comments:

1.In the introduction the authors claim: "This review aims to explore the diverse manifestations of cell-cell interactions in the context of cancer, providing an overview of quantitative approaches to assess these interactions. Moreover, it seeks to investigate the potential of leveraging the influence of cell -cell interactions to enhance the efficacy of cancer immunotherapies." This sounds very general, however, the authors consider only cancer and immune cells as targets for immunotherapy, although others, in particular fibroblasts, could also be considered as potential targets.

Some of the recently published examples:

Targeting cancer associated fibroblasts to enhance immunotherapy: emerging strategies and future perspectives 2021 Jul 6;12(14):1427-1433. doi: 10.18632/oncotarget.27936. 

 Cancer-associated fibroblasts: an emerging target of anti-cancer immunotherapy. J Hematol Oncol 12, 86 (2019). https://doi.org/10.1186/s13045-019-0770-1 

 Roles of cancer-associated fibroblasts (CAFs) in anti- PD-1/PD-L1 immunotherapy for solid cancers. Mol Cancer 22, 29 (2023). https://doi.org/10.1186/s12943-023-01731-z

I think that it would be highly desirable to mention in the introduction and include in the discussion a brief mention of the possibility of other elements of the tumor microenvironment besides the immune cells themselves in the formation of the immune response and their possible use as targets for cancer immunotherapy.

Author Response

Response: this is a great point. We agree that intercellular interactions are not restricted to tumor-immune cell interactions. Numerous other cell types also play roles during immunotherapy. To address this, we have incorporated the following content into the manuscript.

Revision: In the introduction, we added:

“ This resistance is mediated by immunosuppressive cell types, including tumor-associated macrophages (TAMs), myeloid-derived suppressor cells (MDSCs), cancer-associated fibroblast (CAF), and regulatory T cells (Tregs) [10]. These cells actively communicate with other components of the TME, dampening T cell activation, modulating effector cell function, and promoting tumor progression. Of special significance, CAFs and TAMs play pivotal roles in carcinogenesis and the maturation of TMEs [10]. CAFs can promote tumor growth, angiogenesis, invasion of tumor cells into surrounding tissues, and modulating tumor response to immunotherapy [11–13]. They also secrete various signaling molecules and cytokines that can modulate immune responses and create an environment favorable for tumor growth. The equilibrium phase sets the stage for the subsequent escape phase, characterized by clinically detectable tumor growth and the need for therapeutic intervention [9,10].”

Reviewer 2 Report

The review entitles “Decoding the Complexity of Immune-Cancer Cell Interactions: Empowering the Future of Cancer Immunotherapy” underlines the significance of understanding how these cellular communications shape the tumor landscape and influence therapeutic outcomes.  Particularly, the spotlight is cast on the interplay between immune cells and cancer cells, which forms the core of immunotherapy effectiveness and patient-specific responses. The idea that the heterogeneous response to immunotherapy can be attributed to this intricate communication is explored with depth and clarity. The authors do an excellent job researching the arsenal of experimental, statistical, and computational techniques available for dissecting these intercellular communications. The methods, including ligand-receptor analysis, intercellular proximity labeling, and imaging-based approaches, are effectively detailed, showcasing their potential to unravel the disrupted cellular interactions within the TME. Furthermore, the review takes a forward-looking stance by acknowledging the recent advances in imaging, cellular engineering, and bioinformatics as potent tools for uncovering novel cellular relationships. The synergy of these techniques holds the promise of bridging existing knowledge gaps and bolstering the efficacy of cancer immunotherapy.

I have one minor suggestion, as immune cells and cancer cells secrete various cytokines, chemokines, and growth factors that regulate immune responses and inflammation. These signaling molecules can attract immune cells to the tumor site, initiate immune responses, and influence the behavior of surrounding cells. It would be better if the authors addressed this section in their review. 

I enjoy reading this article and recommend accepting this manuscript with minor revision.

Author Response

Response: thank you for the positive comments and we have added a paragraph to address the concern.

Revision: In the section of “Bioinformatic techniques for inferring cell-cell interactions”, we added:

“This approach proves especially valuable in deducing intercellular interactions that aren't solely reliant on cell-to-cell contact. This is evident when immune cells and cancer cells release diverse cytokines, chemokines, and growth factors that govern immune reactions and inflammation. These signaling molecules and their corresponding receptors may be modulated based on environmental cues. Through these bioinformatics methods, we can deduce the likelihood of intercellular interactions based on their ligand-receptor profiles, which is potentially pivotal in forecasting patient prognosis and treatment outcomes.”